# PDED: REVITALIZE PHYSICS LAWS SUBMERGED IN DATA INFORMATION FOR TRAFFIC STATE ESTIMATION

## ABSTRACT

Traditional physics-informed deep learning combines the data-driven methods with the model-based methods by incorporating physics loss as a constraint in total loss function in general, which aims to enforce the neural network to behave according to the physics property. However, this simple integration makes physical knowledge submerged in data information since data loss and physics loss could have large magnitude differences, conflicting directions of the gradients, and varying convergence rates so that the physics law may not work as expected and inhibits the model from working effectively furthermore, especially for traffic state estimation (TSE). To alleviate these issues, we propose a **P**hysical knowledge combined **D**ata information neural network with **E**nsemble **D**istillation framework (PDED) to first disentangle the data-driven model and physics-based model, and then reassemble them to take advantages of label information and physics property. Practically, we separately train data-driven model based on true labels and physics-based model according to physics laws. Then, we introduce the ensemble learning and knowledge distillation to assemble their representations of these two models for constructing a more competitive learnable online teacher model, which in turn distills knowledge to guide the update of them for learning richer knowledge to improve the performance of student models. Through extensive experiments on both synthetic dataset and real-world datasets, our model demonstrates better performance than the existing state-of-the-art methods.

## 1 INTRODUCTION

Traffic state estimation (TSE) is of great significance in traffic management and automated travel field, thus it has attracted the attention of many experts and scholars in recent decades. TSE refers to the data mining problem of reconstructing traffic state variables, including but not limited to flow, density, and velocity, on road segments using partially observed data from traffic sensors (Seo et al., 2017). TSE can be typically divided into two kinds of approaches: model-driven and data-driven (Di & Shi, 2021). The model-driven approach is a mathematical model established based on the prior knowledge of internal physics laws for the development of things. It is a white box model whose parameters have clear physical meanings (Lighthill & Whitham, 1955; Aw et al., 2002). However, model-driven methods require much domain expert knowledge and are usually a simplification of the real situation. On the contrary, the data-driven approach is a black box model that extracts information from complex data, uses deep learning and machine learning models for training and fitting, and finally forms a prediction model. However, it requires a large amount of data for training and optimization to obtain acceptable performance. Moreover, the quality and quantity of real-world data often cannot meet the requirements of training processes, which further affects the model performance.

Recently, researchers have proposed physics-informed deep learning (PIDL) by using physics laws as constraints to aid the training process of data-driven approaches, which is beneficial to enforce the neural network to behave according to physics laws. Thus, PIDL is introduced to TSE (Huang & Agarwal, 2020; Shi et al., 2021b), and then many subsequent studies have expanded the physics models, including integrating macro and micro models (Liu et al., 2021; Barreau et al., 2021), researching second-order models (Shi et al., 2021a) as well as discrete model (Huang & Agarwal, 2022), and introducing the unknown diagram estimation (Shi et al., 2021c). Overall, it can be observed that current researches on PIDL for TSE mainly focus on improving the physical con-

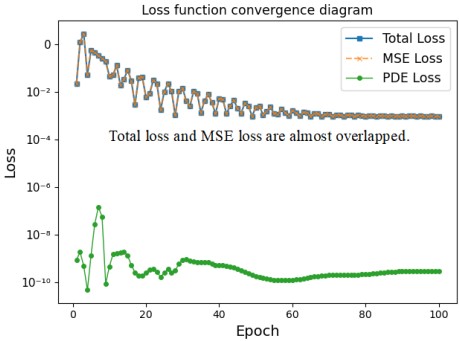 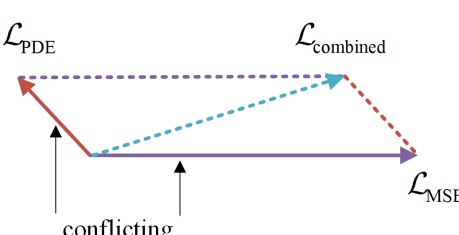

Figure 1: Convergence rates and directions of different loss function. In the left figure, the vertical axis represents the magnitude of loss and the horizontal axis represents the training epoch. In the right figure, $\mathcal{L}_{\mathrm{PDE}}$ and $\mathcal{L}_{\mathrm{MSE}}$ represent the loss based on physics laws and MSE loss, respectively, while $\mathcal{L}_{\mathrm{combined}}$ represents the combined loss.

straint forms, aiming to improve the prediction accuracy of PIDL through more complex traffic flow models. However, these existing researches ignore that there are some inherent limitations in this framework.

In previous PIDL studies, the MSE loss function of predicting traffic states and the partial differential equations (PDE) loss function of traffic states dynamics are combined by addition in a total loss. However, this simple fusing way does not take into account that these two kinds of loss functions could have large magnitude differences, varying convergence rates, and conflicting directions of the gradients. Therefore, directly optimizing the total loss could lead to undesirable performance, since the optimization process struggles to make full use of these two kinds of information so that the model performance is improved limitedly. Take the US-80 dataset as an example in Figure 1, we visualize the convergence rate and directions of two separate loss functions and their combined loss function. The left figure demonstrates that the PDE loss is significantly smaller than the MSE loss since their magnitude difference is about $10^6$, and it exhibits a significantly different convergence rate compared to the MSE loss. Therefore, the PIDL model is predominantly influenced by the MSE loss function, and the physics information is obviously submerged in the data information. We theoretically analyze that the PDE loss has a much smaller magnitude to support the above experimental results in Appendix A.1. The right figure shows the conflicting gradient directions of PDE loss and MSE loss so that directly optimizing the total loss would be detrimental to the model performance. The relative theory analysis can be seen in Appendix A.2.

To solve the above problems, we propose a **P**hysical knowledge combined **D**ata information neural network with **E**nsemble **D**istillation framework (PDED) to disentangle the data-driven model and physics-based model as two separate student models, and then introduce the ensemble learning and knowledge distillation to construct a more competitive teacher model. We first use different student neural networks (NNs) to solve the PDEs of traffic state dynamics and traffic states prediction respectively. Considering the sparse and noisy measurement and complex real-world situation, the performances of the data-driven model and physics-based model are both not satisfied. Therefore, inspired by the idea of ensemble learning (Hansen & Salamon, 1990), we innovatively assemble the feature representations of two student models for constructing a stronger learnable online teacher model, which in turn distills knowledge to guide the update of them for learning richer knowledge and facilitates to optimize a more stable model. In addition, we use Bayesian Neural Networks (BNNs) to quantify the data uncertainty referring to measurement noise in the real world. The obtained uncertainties are utilized to determine the weights of student models in the ensemble process. TSE task only utilizes position and time to predict traffic states without historical traffic states, and the relationships between state variables as PDE describes are beneficial for TSE, which is ignored by previous methods. Therefore, we add the traffic state relation loss to encourage the student to mimic the teacher's state relation matrix. Our contributions are summarized as follows:

- We discover the common limitations of previous PIDL methods that the physics laws are usually submerged in the data information and thus can not work as expected. Therefore,

we propose to first disentangle and then assemble theory to manage physical knowledge and data information.

- We introduce the idea of ensemble learning and knowledge distillation into PDED to assemble the feature representations of two student models to form a more powerful online teacher model, which further distills knowledge back to guide the optimization of student models.

- Considering the data uncertainty in real world, a BNN is utilized to quantify the uncertainty, which is then used as the weight to guide the fusion of the two student models.

- Extensive experiments have been conducted on both synthetic and real datasets, indicating that our framework outperforms the state-of-the-art methods.

## 2 METHODOLOGY

We elaborate on the proposed PDED from three aspects: problem statement, theoretical background and general model framework. The overview of the PDED framework is shown in Figure 2.

### 2.1 PROBLEM STATEMENT

Given a road in the spatial and temporal domain $\mathcal{D} = \{(x, t) | \forall x \in \mathcal{X}, t \in \mathcal{T}\}$, where $x$ represents the position and $t$ represents the time. By measuring the fixed position detectors or moving probe vehicles with a certain frequency, we can observe some traffic state variables, such as velocity $v$ and density $\rho$. We can discretely represent the region using grid points, and thus the whole region is represented as $\mathcal{C} = \{(x_c^{(j)}, t_c^{(j)})\}_{j=1}^{Nc}$. The observed region is represented as $\mathcal{O} = \{(x_o^{(i)}, t_o^{(i)})\}_{i=1}^{N_d}$, and the unobserved region is represented as $\mathcal{U} = \{(x_u^{(i)}, t_u^{(i)})\}_{i=1}^{N_u}$, where $\mathcal{U} = \mathcal{C} \setminus \mathcal{O}$. The task objective is utilizing the observed data points $(x, t)$ in $\mathcal{O}$ rather than historical traffic states to predict the traffic state variables for unobserved region $\mathcal{U}$, which is not a temporal problem essentially. In addition, the observed dataset at different position is sparse and unequal spacing at temporal dimension, so it is improper to use time series models.

### 2.2 THEORETICAL BACKGROUND

In the Introduction Section, we reveal the fact that simple addition of MSE loss and PDE loss is difficult to make full use of their own knowledge in the existing methods. We conduct a mathematical analysis to uncover the reason that the PDE loss has an extremely small magnitude in Appendix A.1 and its gradient directions may conflict with MSE loss in Appendix A.2, which can explain why it hardly works effectively compared to the MSE loss. Naturally, we disentangle the data-driven model and physics-based model as students to fit their own objective functions.

Considering the sparse and noisy measurements as well as complex real-world situations, the performances of the data-driven model and physics-based model may not be satisfied. Then, we propose to assemble their feature representations to improve the model performance. We make theory analyses on the feasibility of regarding the data-driven model and physics-based model as student models to learn a stronger teacher model. These analyses lay solid theoretical evidence for us to propose PDED framework, which effectively integrates the data information and physics laws of PDE.

**Ensemble Learning** To achieve better performance, the data-driven model and physics-based model are treated as student models, and their feature representations are assembled to train a more competitive teacher model. Inspired by the ensemble learning theory (Krogh & Vedelsby, 1994; Zhou et al., 2002), we present the following theorem to show that the performance of ensemble model combined with the data-driven model and physics-based model is determined by two factors.

**Theorem 2.1.** *Let $h_p$ and $h_d$ represent the physical-based model and data-driven model respectively, $H$ represents the ensemble teacher model. Then the ensemble generalization error $E$ of $H$ is defined as the weighted sum of generalization errors for individual models subtracting the differences between the physical-based model and the data-driven model.*

$$E = \overline{E} - \overline{A} \tag{1}$$

where $E$ denotes the ensemble generalization error, $\overline{E}$ denotes the weighted sum of generalization error for $h_p$ and $h_d$, $\overline{A}$ denotes the weighted sum of individual model differences.

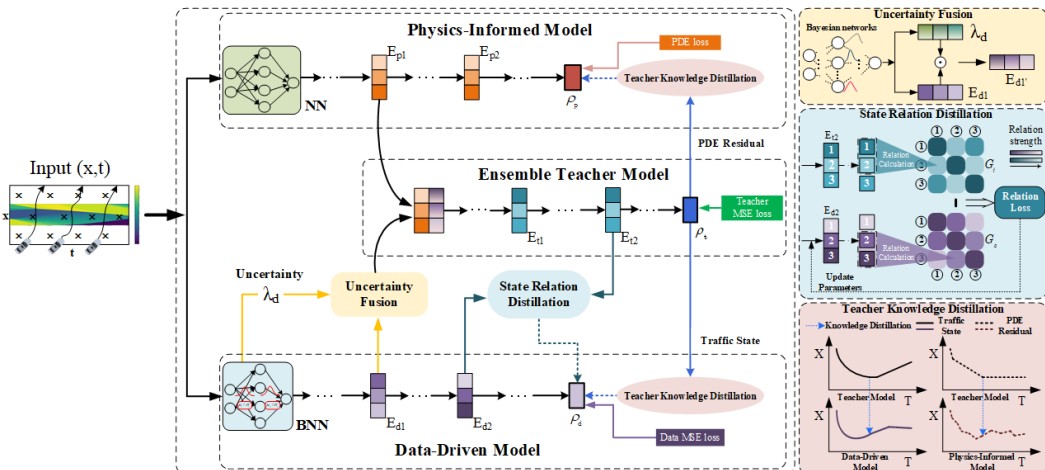

Figure 2: The overview of proposed PDED framework that consists of three models: Physics-Informed Model, Data-Driven Model and Ensemble Teacher Model.

The theorem indicates that the higher the accuracy of student models, and the greater the diversity between the student models, the better the ensemble model performance is. Obviously, the physical-based model and data-driven model are totally different, since they optimize the model from different aspects: the physical-based model learns the intrinsic physics laws while the data-driven model updates parameters from labeled data. Thus, assembling the physical-based model and data-driven model would be effective from theory analysis. Moreover, we use ensemble teacher to distill knowledge back to student models to improve their accuracy. The detailed proof of Theorem 2.1 can be seen in Appendix A.3.

### 2.3 PDED: Proposed Framework

In this section, we describe the proposed PDED for traffic states estimation, which mainly consists of three models: the physics-informed model, the data-driven model and the ensemble teacher model. The physics-informed model is responsible for encoding traffic-related physics laws, while the data-driven model learns information from the labeled measurements. These two models are disentangled as student models, whose embeddings are then integrated to construct a more competitive teacher model, which in turn distills knowledge to the student models to improve their performance. We also introduce Bayesian Neural Networks (BNN) to handle data uncertainty, and use traffic state relation as supervision to improve the model accuracy.

#### 2.3.1 Physics-informed Model

The student physics-informed model $S_p$ is proposed to learn physics laws in traffic state estimation. Here, we utilize two most classical models: the first-order Lighthill-Whitham-Richards (LWR) model (Lighthill & Whitham, 1955) and the second-order Aw-Rascle-Zhang (ARZ) model (Aw et al., 2002). The LWR model provides the relationship between traffic flow and traffic density on the road, which is described as follows:

$$\rho_t + (q(\rho))_x = 0, \quad q(\rho) = \rho v_f \left(1 - \frac{\rho}{\rho_m}\right). \tag{2}$$

where $q$ represents flow and $\rho$ represents density, $\rho_t$ is the partial differential of $\rho$ with respect to $t$, $(q(\rho))_x$ is the partial differential of $q$ with respect to $x$, $v_f$ and $\rho_m$ are the maximum velocity and density, respectively. Compared to the LWR model, the second-order ARZ model contains both a conservation law of vehicles and a momentum equation on velocity. The detailed ARZ model equations can be seen in Appendix B.1.

We use the neural network to predict the traffic states by the input $(x, t)$, which should obey the PDE of traffic state dynamics.

$$\rho_p = \text{NN}(x, t, \theta_p), \tag{3}$$

where $\rho_p$ denotes the predicted traffic states by the physics-informed model, $\theta_p$ denotes the model parameters. The physics-informed model is trained by PDE loss as follows:

$$\mathcal{L}_{pde-lwr} = \frac{1}{N_p} \sum_{i=1}^{N_p} \left| f_{lwr}(x^i, t^i, \theta_p) \right|^2, \, f_{lwr}(x, t, \theta_p) = v_f \left( 1 - \frac{2\rho_p}{\rho_m} \right)(\rho_p)_x + (\rho_p)_t. \quad (4)$$

where $f_{lwr}(x, t, \theta_p)$ denotes the residual value of PDE based on LWR, $N_p$ is the number of auxiliary points. Eq.( 4) corresponds to the loss functions of the LWR model. By minimizing the loss of Eq.( 4), the optimal solution for the physics-informed model can be obtained. The loss function of the ARZ model can be seen in Appendix B.1.

### 2.3.2 DATA-DRIVEN MODEL

The student data-driven model $S_d$ is utilized to learn information from real-world labeled data. Since the traffic states measurements may have uncertainty, we use a BNN to model the uncertainty:

$$\mathbf{E}_{d_1} = \text{BNN}(x, t, \theta_b), \quad (5)$$

where $\theta_b$ denotes the model parameters of BNN. Subsequently, we use a multi-layer perceptron operation to obtain the embedding $\mathbf{E}_{d_2} = \text{MLP}(\mathbf{E}_{d_1})$ aiming to calculate the traffic state relations in the following section. Finally, a linear mapping is applied to acquire the final prediction result $\rho_d$.

$$\rho_d = \mathbf{E}_{d_2} \mathbf{W}_d + \mathbf{b}_d \quad (6)$$

We train the data-driven model using real-world labeled data by the following loss.

$$\mathcal{L}_{mse-d} = \frac{1}{N_d} \sum_{i=1}^{N_d} \left| \rho_d^i - \rho_{true}^i \right|^2 \quad (7)$$

where $\rho_{true}$ is the measurement traffic states, $N_d$ is the sample number.

### 2.3.3 ENSEMBLE TEACHER MODEL

Single data-driven model or physics-based model can not get the satisfied performance, thus we assemble their embeddings to construct a more powerful learnable teacher model. We first integrate the student representations of $S_p$ and $S_d$ with consideration of the data uncertainty quantified by $S_d$ to get the input $\mathbf{E}_{t_1}$ for ensemble teacher model.

$$\mathbf{E}_{t_1} = (\mathbf{E}_{p_1} + \lambda_d \mathbf{E}_{d_1}) \mathbf{W}_t + \mathbf{b}_t \quad (8)$$

where $\lambda_d$ is the uncertainty score that is calculated based on the posterior covariance $\mathbf{C}$ of the BNN in $S_d$: $\lambda_d = \text{sigmoid}(\mathbf{C}\mathbf{W}_c + \mathbf{b}_c)$. $\mathbf{E}_{p_1}$ is the first layer output of $S_p$, $\mathbf{E}_{d_1}$ is the output of BNN. Then, we use an additional neural network to construct a learnable stronger online teacher. Finally, the predicted traffic states of teacher model is calculated as follows:

$$\rho_t = \text{NN}(\mathbf{E}_{t_1}, \theta_t) \quad (9)$$

where $\theta_t$ is the parameters of teacher model. We train the ensemble teacher model based on the following MSE loss.

$$\mathcal{L}_{mse-t} = \frac{1}{N_d} \sum_{i=1}^{N_d} \left| \rho_t^i - \rho_{true}^i \right|^2 \quad (10)$$

The teacher model contains both physical knowledge and the labeled data information so that it would have better performance. As mentioned by Theorem 2.1 , the effectiveness of ensemble teacher model is also determined by the the accuracy of individual student model. Therefore, we will use the teacher model to distill knowledge back to guide the update of student model for learning richer knowledge to improve the performance of individual student.

**Physical Knowledge Distillation**  We use teacher model to guide the student model $S_p$ to learn the physics laws of traffic dynamics with teacher distillation PDE loss $\mathcal{L}_{pde-st}$.

$$\mathcal{L}_{pde-st} = \frac{1}{N_p} \sum_{i=1}^{N_p} \left| f_s(x^i, t^i, \theta_p) - f_t(x^i, t^i, \theta_t) \right|^2 \quad (11)$$

where $f_s(x, t, \theta_p)$ is the residual value of PDE for student model $S_p$, $f_t(x, t, \theta_t)$ is the residual value of PDE for teacher model. $f_s(x, t, \theta_p)$ and $f_t(x, t, \theta_t)$ can be either residual value of LWR or ARZ, but should be the same.

**Data Information Distillation** On the other hand, we use teacher model to guide the student model $S_d$ to update the model parameters in order to transfer knowledge from a stronger teacher to the student $S_d$ with teacher distillation MSE loss $\mathcal{L}_{mse-st}$.

$$\mathcal{L}_{mse-st} = \frac{1}{N_d} \sum_{i=1}^{N_d} \left| \rho_d^i - \rho_t^i \right|^2 \tag{12}$$

**Traffic State Relation Distillation** Previous methods ignore the interdependence among traffic state variables. In fact, the relationships between state variables have a positive effect on predicting states at different times and positions (Tung & Mori, 2019). Adding supervision signals in intermediate layers can effectively enhance the supervisory information from teacher model to student model. Thus, we use the teacher distillation relation loss $\mathcal{L}_r$ to distill the knowledge from teacher to data-driven model $S_d$.

$$\mathcal{L}_r = \frac{1}{N_d^2} ||\mathbf{G}_t - \mathbf{G}_s||_F^2 \ , \tag{13}$$

where $|| \cdot ||_F$ is the Frobenius norm, $\mathbf{G}_t$ and $\mathbf{G}_s$ are traffic state relation matrices of teacher model and student model $S_d$, whose calculation process can be seen in Appendix B.2.

### 2.3.4 MODEL TRAINING

According to the above analysis, the loss function of physics-informed model $S_p$ and data-driven model $S_d$ are summarized as follows:

$$\mathcal{L}_{phy} = \alpha_1 \mathcal{L}_{pde} + \alpha_2 \mathcal{L}_{pde-st}, \tag{14}$$
$$\mathcal{L}_{data} = \beta_1 \mathcal{L}_{mse-d} + \beta_2 \mathcal{L}_{mse-st} + \beta_3 \mathcal{L}_r. \tag{15}$$

where $\alpha_1$, $\alpha_2$, $\beta_1$, $\beta_2$ and $\beta_3$ are hyper-parameters to balance how each component influences the total loss. It is important to note that each component loss has relatively same magnitude in $\mathcal{L}_{phy}$ or $\mathcal{L}_{data}$, and thus it is quite easy for $S_p$ and $S_d$ to optimize parameters to obtain the best performance. The pseudo-code of our PDED framework is summarized in Appendix B.3

## 3 EXPERIMENT

### 3.1 DATASETS AND SETTINGS

**Datasets** To demonstrate the effectiveness of our model, we conduct sufficient experiments on one synthetic dataset and two real datasets. The learning traffic states are the velocity $v(x,t)$ and density $\rho(x,t)$. (1) **Synthetic dateset**: As mentioned in previous methods (Huang & Agarwal, 2022), we use synthetic dataset constructed by the Lax-Hopf method (Mazare et al., 2011). The synthetic dataset is a 5000-meter road segment for 300 seconds ($x, t \in [0, 5000] \times [0, 300]$), the spatial resolution of the dataset is 5 meters and the temporal resolution is 1 second ($\Delta x = 5$, $\Delta t = 1$). (2) **US-80**: It is the vehicle trajectory data collected on the interstate 80 freeway in Emeryville, California of Next Generation Simulation (NGSIM) dataset (of Transportation, 2008). (3) **US-101**: It is collected on highway US 101 of the NGSIM dataset.

**Baselines** We compare PDED with the widely used traffic state estimation models, including (1) **NN**: the pure neural network model; (2) **EKF** (Seo & Bayen, 2017): the ARZ-based extended Kalman filter (EKF) which is a representative model-based TSE method. (3) **PIDL-LWR** (Liu et al., 2021; Huang & Agarwal, 2020): the neural network regarding LWR model as constraints; (4) **PIDL-ARZ** (Shi et al., 2021a): the neural network regarding ARZ model as constraints.

**Experimental Settings** We conduct the experiments under different penetration rates (P-Rate) for synthetic dateset, and different proportions of loop detectors for US-80 and US-101 as training data. Weight coefficients of losses $\alpha_1$, $\alpha_2$, $\beta_1$, $\beta_2$ and $\beta_3$ are all set to 1. To achieve optimal performance of GLIM, the hyper-parameters are determined by a grid search during the training process. As the losses of the data-driven model and ensemble teacher are related to real labeled data, we set their learning rates to the same value, varying among {0.01, 0.005, 0.001, 0.0005}. The loss of the physics-informed model is much smaller, so we vary its learning rate from {0.01, 0.005, 0.001, 0.0005. 0.0001}. The layer numbers of the three models are all set to 3. Adam (Kingma & Ba, 2015) is used to optimize the parameters in our model. We use the common metric $L^2$ relative error to quantify the estimation error (Shi et al., 2021a). We test the experimental results of our framework with two variants PDED-LWR and PDED-ARZ.

Table 1: Overall performance of Synthetic dataset (Error). P-Rate: Penetration rate. Bold: best

| Traffic-State | P-Rate/N-car | NN | EKF | PIDL-LWR | PIDL-ARZ | PDED-LWR | PDED-ARZ |
|---|---|---|---|---|---|---|---|
| $v$ | 1%:4 | 0.3873 | 0.3520 | 0.3580 | 0.3652 | **0.3254** | 0.3274 |
| | 10%:40 | 0.3368 | 0.3281 | 0.2062 | 0.1822 | **0.1225** | 0.1287 |
| | 30%:120 | 0.1867 | 0.3214 | 0.1527 | 0.1603 | **0.1167** | 0.1201 |
| | 50%:200 | 0.1589 | 0.3202 | 0.1321 | 0.1397 | **0.0948** | 0.1107 |
| | 70%:280 | 0.1475 | 0.3148 | 0.1104 | 0.1197 | **0.0916** | 0.1095 |
| $\rho$ | 1%:4 | 0.3642 | 0.3197 | 0.3291 | 0.3351 | **0.2278** | 0.2491 |
| | 10%:40 | 0.3097 | 0.3011 | 0.1841 | 0.2202 | **0.1019** | 0.1205 |
| | 30%:120 | 0.1767 | 0.2984 | 0.1408 | 0.1644 | **0.0917** | 0.1065 |
| | 50%:200 | 0.1408 | 0.2934 | 0.1184 | 0.1271 | **0.0821** | 0.0845 |
| | 70%:280 | 0.1102 | 0.2888 | 0.0912 | 0.0985 | **0.0805** | 0.0835 |

## 3.2 Overall Performance

No matter for synthetic or real datasets, our proposed PDED outperforms other baselines across different penetration rates and different proportions of loop detectors, which proves the rationality of our ensemble distillation framework. By disentangling MSE loss and PDE loss and adopting ensemble distillation, we effectively avoid the direct addition of two obviously different losses in traditional PIDL methods, leading to a significant improvement in model performance.

Comparing the performance of different types of baselines, we can find that in the low training data, the purely data-driven approach NN performs worst. However, as the training data increases, the effectiveness of NN improves rapidly. In contrast, the model-based approach EKF demonstrates its superiority when the training data is extremely low (1%), second only to our PDED model. As the training data increases, the performance of EKF improves limitedly. When above 10% training data, the NN outperforms the EKF for both $\rho$ or $v$. The PIDL-based methods combine the advantages of data-driven and model-based approaches, thus they outperform single data-driven or model-based methods with 30% training data or higher.

The LWR variants are superior to ARZ variants in both synthetic and real datasets, which is beyond our expectations as the ARZ model introduces the momentum equation on velocity compared to the LWR model. For synthetic dataset, we find that the synthetic dataset generated through the Lax-Hopf method is based on the LWR model. Therefore, the LWR variants have better prediction results for the synthetic dataset. For real datasets, the results prove that sophisticated traffic models may not always lead to better performance as the model may contain more complicated terms that make the TSE performance more sensitive, and thus ARZ variants could not perform well in some specific situations.

## 3.3 Ablation Study

To evaluate the contributions of each component of our PDED framework, we further compare the full PDED framework with those variants: (1) w/o $\mathcal{L}_r$ denotes PDED without teacher distillation relation loss; (2) w/o BNN denotes that PDED use NN instead of BNN in data-driven model; (3) w/o $\mathcal{L}_{mse-st}$ denotes PDED without teacher distillation MSE loss; (4) w/o $\mathcal{L}_{pde-st}$ denotes PDED without teacher distillation PDE loss; (5) w/o $\mathcal{L}_{mse-st} + \mathcal{L}_{pde-st}$ denotes PDED without both teacher distillation MSE loss and teacher distillation PDE loss.

Due to limited space, we only present the results of the ablation study of PDED-LWR with 30% data in Table 3. More ablation experiments can be seen in the Appendix C.2. We can observe that (1) the full PDED achieves the best performance, which indicates each component of the model is beneficial to the overall performance. (2) The ensemble distillation framework plays the core role in the proposed PDED. Either removing the distillation loss in the data-driven model ($\mathcal{L}_{mse-st}$) or in the physics information model ($\mathcal{L}_{pde-st}$), the performance has significantly reduced. When both losses are removed, the model gets the worst performance, which proves that using an ensemble teacher to guide student models can improve the effectiveness of student models. (3) w/o $\mathcal{L}_r$ underperforms the full model, which shows the relationships between traffic state variables are helpful for assisting model improvement by utilizing the supervisory information of the intermediate process.

Table 2: Overall performance of real datasets (Error). Bold: best

| Dataset | Traffic State | #Loop | NN | EKF | PIDL-LWR | PIDL-ARZ | PDED-LWR | PDED-ARZ |
|---|---|---|---|---|---|---|---|---|
| US-80 | $v$ | 8 (10%) | 0.2569 | 0.1853 | 0.1903 | 0.2048 | **0.1245** | 0.1317 |
| | | 24 (30%) | 0.2226 | 0.1816 | 0.1741 | 0.1804 | **0.0956** | 0.1015 |
| | | 40 (50%) | 0.1569 | 0.1767 | 0.1342 | 0.1442 | **0.0882** | 0.0944 |
| | | 56 (70%) | 0.1283 | 0.1713 | 0.1107 | 0.1197 | **0.0874** | 0.0895 |
| | | 72 (90%) | 0.1042 | 0.1687 | 0.0997 | 0.0999 | **0.0861** | 0.0862 |
| | $\rho$ | 8 (10%) | 0.3048 | 0.2525 | 0.2583 | 0.2888 | **0.1906** | 0.2048 |
| | | 24 (30%) | 0.2647 | 0.2501 | 0.2378 | 0.2449 | **0.1789** | 0.1902 |
| | | 40 (50%) | 0.2423 | 0.2486 | 0.2157 | 0.2382 | **0.1717** | 0.1877 |
| | | 56 (70%) | 0.2339 | 0.2454 | 0.2082 | 0.2251 | **0.1653** | 0.1824 |
| | | 72 (90%) | 0.2107 | 0.2442 | 0.2014 | 0.2072 | **0.1622** | 0.1815 |
| US-101 | $v$ | 2 (10%) | 0.1955 | 0.1502 | 0.1517 | 0.1606 | **0.1237** | 0.1385 |
| | | 6 (30%) | 0.1732 | 0.1417 | 0.0932 | 0.1142 | **0.0531** | 0.0705 |
| | | 10 (50%) | 0.1285 | 0.1384 | 0.0798 | 0.0842 | **0.0417** | 0.0646 |
| | | 15 (70%) | 0.0824 | 0.1305 | 0.0685 | 0.0706 | **0.0384** | 0.0604 |
| | | 18 (90%) | 0.0683 | 0.1296 | 0.0607 | 0.0614 | **0.0348** | 0.0599 |
| | $\rho$ | 2 (10%) | 0.2831 | 0.2341 | 0.2376 | 0.2531 | **0.2045** | 0.2242 |
| | | 6 (30%) | 0.2475 | 0.2283 | 0.2224 | 0.2352 | **0.1724** | 0.1932 |
| | | 10 (50%) | 0.2142 | 0.2268 | 0.2039 | 0.2131 | **0.1572** | 0.1875 |
| | | 15 (70%) | 0.2075 | 0.2204 | 0.1942 | 0.2004 | **0.1511** | 0.1823 |
| | | 18 (90%) | 0.1975 | 0.2198 | 0.1865 | 0.1895 | **0.1445** | 0.1783 |

Table 3: Ablation study of PDED-LWR with 30% data. (Error). Bold: best

| Models | US-101 | | US-80 | | Synthetic | |
|---|---|---|---|---|---|---|
| | $v$ | $\rho$ | $v$ | $\rho$ | $v$ | $\rho$ |
| w/o $\mathcal{L}_r$ | 0.0588 | 0.1764 | 0.1062 | 0.1804 | 0.1218 | 0.0984 |
| w/o BNN | 0.0615 | 0.1788 | 0.1128 | 0.1883 | 0.1287 | 0.1008 |
| w/o $\mathcal{L}_{mse-st}$ | 0.0644 | 0.1833 | 0.1192 | 0.1934 | 0.1302 | 0.1057 |
| w/o $\mathcal{L}_{pde-st}$ | 0.0698 | 0.1873 | 0.1225 | 0.1953 | 0.1316 | 0.1132 |
| w/o $\mathcal{L}_{mse-st} + \mathcal{L}_{pde-st}$ | 0.0712 | 0.2004 | 0.1385 | 0.2136 | 0.1376 | 0.1207 |
| PDED-LWR | **0.0531** | **0.1724** | **0.0956** | **0.1789** | **0.1167** | **0.0917** |

(4) Removing BNN also reduces the model performance since BNN can quantify the uncertainty arising from the noisy data to obtain more accurate traffic state predictions in different scenarios.

## 3.4 Noise Robustness Analysis

We conduct experiments on a 30% penetration rate in the synthetic dataset. The noise data are generated by adding a white Gaussian noise $\epsilon \sim \mathcal{N}(0, 0.01)$ to the training synthetic data with different data noise ratios among $\{0.05, 0.1, 0.15, 0.2, 0.25\}$. According to Figure 3, we can see that our PDED framework always achieves the optimal performance for different noise ratios, and as the noise ratio increases, the prediction error increases slowly. However, for NN and PIDL models, when the noise rate increases to a certain proportion, the prediction error shows a turning point that transitions from a slow increase to a rapid increase. This indicates that our framework has better noise robustness compared to the baselines since our framework utilizes the BNN to measure data uncertainty and adopts uncertainty to guide the fusion process. Therefore, the weights of representation learned from data-driven model would be adjusted when data uncertainty increases so that it reduces the influence of the noise data on model training. We also do more experiments for white Gaussian noise $\epsilon \sim \mathcal{N}(0, 0.04)$ in Appendix C.3.

We conduct the parameters sensitivity analysis in Appendix C.4, and we visualize the predictions of the traffic velocity $v$ and traffic density $\rho$ by NN and proposed PDED for different datasets in Appendix C.5.

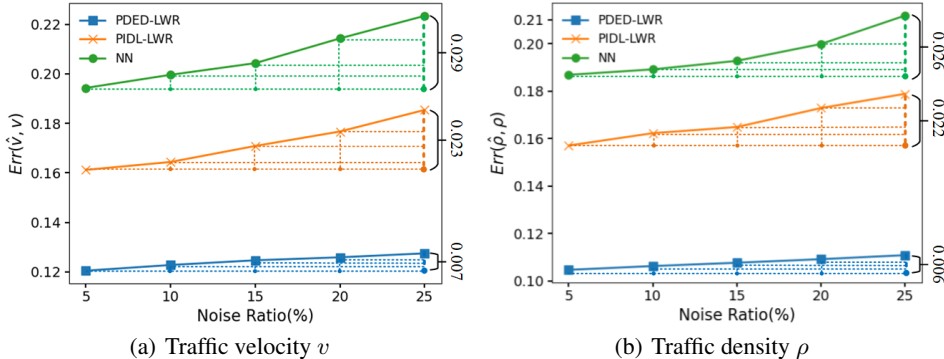

(a) Traffic velocity $v$    (b) Traffic density $\rho$

Figure 3: Noise robustness analysis of white Gaussian noise with variance 0.01. PDED has much smaller increase of predcition error compared to the baselines.

## 4    RELATED WORK

### 4.1    TRAFFIC STATE ESTIMATION

Two types of traditional TSE approaches are model-driven and data-driven (Di & Shi, 2021). Common model-based approaches used in the field of traffic state estimation include the first-order models such as the Lighthill-Whitham-Richards (LWR) (Lighthill & Whitham, 1955; Richards, 1956) model as well as its differentiated version Cell Transmission model (Huang & Agarwal, 2022), and the second-order models such as the Payne Whitham (PW) (Whitham, 1974) model, Aw-Rascle-Zhang (ARZ) (Aw et al., 2002; Zhang, 2002) model and the ARZ-based extended Kalman filter (EKF) (Seo & Bayen, 2017). Data-based approaches automatically extract features from raw data and capture complex temporal and spatial patterns in traffic data (Raissi, 2018; Raissi & Karniadakis, 2018; Tang et al., 2021).

To combine the advantages between these two kinds of approaches, the early PIDL frameworks for TSE are proposed by Huang & Agarwal (2020) and Shi et al. (2021b), which convert the first-order LWR model to the PDE loss. Subsequently, the PDE loss is extended to the second-order ARZ model (Shi et al., 2021a) and neural network is used to estimate hidden variable relationships (Shi et al., 2021c). Other scholars integrate microscopic models such as Follow-the-Leader dynamics (FL1) (Liu et al., 2021; Barreau et al., 2021). Some researches extend to the discrete space CTM (Huang & Agarwal, 2022). Moreover, GAN is utilized to handle data uncertainty (Mo et al., 2022).

### 4.2    DISTILLATION LEARNING

As a representative type of model compression and acceleration, knowledge distillation effectively learns a small student model from a large teacher model (Gou et al., 2021). Knowledge distillation can be divided into two categories: offline distillation (Hinton et al., 2015) and online distillation (Zhang et al., 2018). Ensemble distillation is a unique approach of online distillation as teacher model is a combination of multiple student sub-models. ONE (Zhu et al., 2018) trains a single multi-branch network and uses a gate to ensemble all the branches while simultaneously establishing a strong teacher. KDCL (Guo et al., 2020) designs multiple methods to generate soft targets as supervisions by effectively ensembling predictions of students. PCL (Wu & Gong, 2021) exploits the temporal mean model of each peer as the peer mean teacher to distill knowledge among peers.

## 5    CONCLUSION

In this paper, we propose a novel physical knowledge combined data information neural network with ensemble distillation framework to solve traffic state estimation problems. We separately treat the physics-informed model and data-driven model as student models to avoid adding MSE loss and PDE loss in one model. Furthermore, we introduce ensemble learning and knowledge distillation to form a more powerful online teacher mode, which can in turn guide the optimization of student models. The proposed framework demonstrates its superiority on one synthetic dataset and two real data datasets. Our work provides novel insights into how neural networks can better integrate physical knowledge for future PIDL research.

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

APPENDIX

This appendix provides details on the proof of theorems, our PDED framework, and supplementary experiments.

## A    PROOF OF THEOREMS

### A.1    THE MAGNITUDE OF PDE LOSS

We will explain that the PDE loss has an extremely small magnitude. Take LWR model (Lighthill & Whitham, 1955) as an example, which is a most classical macroscopic first-order traffic flow model expressed in Eq.(2). It describes the relationship between traffic flow and density, and can be used to predict the traffic states on the road.

**Theorem A.1.** *Using a neural network with $L$ layers to predict the traffic density state, the PDE loss of LWR is determined by the parameters of the neural network, and its magnitude is positive correlated with $L$.*

$$L_{phy} = \sum_{j=1}^{N_p} \left| f(x^j, t^j, \theta) \right|^2,$$

$$f(x, t, \theta) = v_f(1 - 2\rho_L) \prod_{n=1}^{L}(1 - \rho_n^2) \prod_{n=2}^{L} W_n W_{11} + \prod_{n=1}^{L}(1 - \rho_n^2) \prod_{n=2}^{L} W_n W_{12}.$$

$$(16)$$

where $\theta, W$ are the parameters of neural network, $\rho_n$ represents the output of the $n$-th layer of the neural network, $\rho_L$ represents the final predicted traffic density state, $f(x, t, \theta)$ represents the residual value. Then, we will describe the calculation process of the residual value $f(x, t, \theta)$. LWR is described in Eq.( 2), and we can combine them as:

$$v_f\left(1 - \frac{2\rho}{\rho_m}\right)\rho_x + \rho_t = 0, \qquad (17)$$

The $\rho_m$ is usually set to be 1 for convenience, and thus the residual value of LWR model is:

$$f(x, t, \theta) = v_f(1 - 2\rho)\rho_x + \rho_t, \qquad (18)$$

Assuming there is only one hidden layer in the neural network, then the output is:

$$\rho = \sigma(W_{11}x + W_{12}t + b_1), \sigma = \tanh(\cdot), \qquad (19)$$

Considering Eq.(18) and Eq.(19), we can convert the residual value into:

$$
\begin{aligned}
f(x, t, \theta) &= v_f(1 - 2\rho)\sigma' W_{11} + \sigma' W_{12} \\
&= v_f(1 - 2\rho)(1 - \rho^2)W_{11} + (1 - \rho^2)W_{12},
\end{aligned}
\qquad (20)
$$

Except for the constant term $v_f$, the absolute value of each term in equation is less than 1. As $\rho \in [0, \rho_m]$ and $\rho_m = 1$, so $1 - \rho^2 \in [0, 1]$ and $|1 - 2\rho| \in [0, 1]$, parameters $|W_{11}|, |W_{12}| \in [0, 1]$ in neural networks. In practice, in order to accurately describe the physics-informed model in PIDL, multi-layer neural network is required, where we set the layer of neural network $L$. We use $\rho_n$ to represent the output of the $n$-th layer in the neural network.

$$\rho_n = \sigma(W_n\rho_{n-1} + b_n). \qquad (21)$$

Finally, we obtain the residual value in Eq.(16). As the researchers usually utilize multi-layer neural network to estimate the traffic density states, the PDE loss is obtained by continuous multiplication of multiple objects, whose values are all smaller than 1, and thus the magnitude of the PDE loss is extremely small compared to the MSE loss. The PDE loss with a small magnitude could not contribute to the optimization process for updating the model parameters so the physical laws do not work as expected. Moreover, it can be found that we use tanh as the activation function, which can be replaced with other smooth nonlinear activation functions, such as sigmoid, and we can get the similar conclusion. As shown in (Mishra & Molinaro, 2022; Markidis, 2021), physics-informed neural network requires sufficient smoothness for the activation function. However, ReLU is only Lipschitz continuous, and thus ReLU and other non-smooth activation functions, such as ELU and Leaky ReLU are not consistent methods, which means the solution does not converge to the exact solution. Therefore, we do not use ReLU-like activation functions in our method.

A.2   THE GRADIENT DIRECTIONS OF PDE LOSS AND MSE LOSS

In the existing methods, the PDE loss and MSE loss are calculated as follows:

$$\mathcal{L}_p = \frac{1}{N_p} \sum_{i=1}^{N_p} \left| f(x^i, t^i, \theta_p) \right|^2 = \frac{1}{N_p} \sum_{i=1}^{N_p} \left| v_f(1-2\rho)(1-\rho^2)W_{11} + (1-\rho^2)W_{12} \right|^2_{\rho=\rho^i}, \quad (22)$$

$$\mathcal{L}_m = \frac{1}{N_d} \sum_{i=1}^{N_d} \left| \rho^i - \rho^i_{true} \right|^2, \quad (23)$$

We just use one hidden layer of neural network to predict the traffic density in Eq.(19) for simplification. Thus the gradients of the PDE loss and MSE loss with respect to parameters $W_{11}$ are:

$$\frac{\partial \mathcal{L}_p}{\partial W_{11}} = \frac{2}{N_p} \sum_{i=1}^{N_p} \left( v_f(1-2\rho)(1-\rho^2)W_{11} + (1-\rho^2)W_{12} \right) * \left( v_f(1-2\rho)(1-\rho^2) \right) |_{\rho=\rho^i}$$

$$\frac{\partial \mathcal{L}_m}{\partial W_{11}} = \frac{2}{N_d} \sum_{i=1}^{N_d} \left( \rho^i - \rho^i_{true} \right) x \quad (24)$$

The gradients of the PDE loss and MSE loss with respect to parameters $W_{12}$ are:

$$\frac{\partial \mathcal{L}_p}{\partial W_{12}} = \frac{2}{N_p} \sum_{i=1}^{N_p} \left( v_f(1-2\rho)(1-\rho^2)W_{11} + (1-\rho^2)W_{12} \right) * (1-\rho^2) |_{\rho=\rho^i}$$

$$\frac{\partial \mathcal{L}_m}{\partial W_{12}} = \frac{2}{N_d} \sum_{i=1}^{N_d} \left( \rho^i - \rho^i_{true} \right) t \quad (25)$$

$\frac{\partial \mathcal{L}_m}{\partial W_{11}}$ and $\frac{\partial \mathcal{L}_m}{\partial W_{12}}$ must have the same sign, since $x > 0, t > 0$. When $\rho > 0.5$, $1 - 2\rho < 0$, and thus $\frac{\partial \mathcal{L}_p}{\partial W_{11}}$ and $\frac{\partial \mathcal{L}_p}{\partial W_{12}}$ must have the different signs. That means either the gradients with respect to parameters $W_{11}$ between the PDE loss and MSE loss or the gradients with respect to parameters $W_{12}$ between the PDE loss and MSE loss must be conflicting. When $\rho < 0.5$, $\frac{\partial \mathcal{L}_p}{\partial W_{11}} > 0$ and $\frac{\partial \mathcal{L}_p}{\partial W_{12}} > 0$. The signs of $\frac{\partial \mathcal{L}_m}{\partial W_{11}}$ and $\frac{\partial \mathcal{L}_m}{\partial W_{12}}$ depend on the model prediction and the distribution of traffic density. Assuming the distribution of traffic density is Gaussian distribution and $\frac{\partial \mathcal{L}_m}{\partial W_{11}}$ and $\frac{\partial \mathcal{L}_m}{\partial W_{12}}$ being positive or negative is fifty-fifty. In conclusion, the gradient directions of PDE loss and MSE loss are most likely conflicting.

A.3   ENSEMBLE LEARNING

Let $h_p$ and $h_d$ represent the physical-based model and data-driven model respectively. $H = w_p h_p + w_d h_d$ represents the ensemble teacher model, the difference between a single student model and ensemble teacher model is:

$$A(h_p|x) = (h_p(x) - H(x))^2,$$
$$A(h_d|x) = (h_d(x) - H(x))^2, \quad (26)$$

The overall **difference** between ensemble teacher model and all student models is:

$$\overline{A}(h|x) = w_p(h_p(x) - H(x))^2 + w_d(h_d(x) - H(x))^2, \quad (27)$$

Let $f(x)$ be the ground truth, the **errors** of individual student models and ensemble teacher model are:

$$E(h_p|x) = (f(x) - h_p(x))^2,$$
$$E(h_d|x) = (f(x) - h_d(x))^2, \quad (28)$$
$$E(H|x) = (f(x) - H(x))^2,$$

The weighted sum of individual student model error is:

$$\overline{E}(h|x) = w_p E(h_p|x) + w_d E(h_d|x),$$
$$w_p + w_d = 1, \quad (29)$$

Based on the definition of difference and error in Eq.(27) and Eq.(28), the difference $\overline{A}(h|x)$ between physics-based model and data-driven model can be deduced as follow:

$$
\begin{aligned}
\overline{A}(h|x) &= w_p(h_p(x) - H(x))^2 + w_d(h_d(x) - H(x))^2 \\
&= w_p(h_p(x)^2 - 2h_p(x)H(x) + H(x)^2) + w_d(h_d(x)^2 - 2h_d(x)H(x) + H(x)^2) \\
&= w_p h_p(x)^2 + w_d h_d(x)^2 - H(x)^2 \\
&= w_p(f(x) - h_p(x))^2 + w_d(f(x) - h_d(x))^2 - (f(x) - H(x))^2 \\
&= w_p E(h_p|x) + w_d E(h_d|x) - E(H|x),
\end{aligned}
\tag{30}
$$

Since the above equation holds for all samples $x$, let $p(x)$ represents the probability density of the sample, then the difference for all samples satisfies:

$$
\begin{aligned}
&w_p \int A(h_p|x)p(x)dx + w_d \int A(h_d|x)p(x)dx \\
&= w_p \int E(h_p|x)p(x)dx + w_d \int E(h_d|x)p(x)dx - \int E(H|x)p(x)dx,
\end{aligned}
\tag{31}
$$

Set the weighted sum of individual student model differences as $\overline{A} = w_p \int A(h_p|x)p(x)dx + w_d \int A(h_d|x)p(x)dx$, the weighted sum of generalization error for $h_p$ and $h_d$ as $\overline{E} = w_p \int E(h_p|x)p(x)dx + w_d \int E(h_d|x)p(x)dx$, and the ensemble generalization error as $E = \int E(H|x)p(x)dx$. Then, we can obtain:

$$
E = \overline{E} - \overline{A}.
\tag{32}
$$

The beauty of this theorem is that it separates the generalization error into a term that depends on the generalization errors of the individual models and another term that contains all correlations between the individual models. The theorem indicates that the higher the accuracy of individual models, and the greater the diversity between the individual models, the better the ensemble model performance is. Obviously, the physical-based model and data-driven model are totally different, since they optimize the model from different aspects: the physical-based model learns the intrinsic physics laws while the data-driven model updates parameters from labeled data. Thus, assembling the physical-based model and data-driven model would be effective from theory analysis. Moreover, we will distill knowledge from the stronger teacher to each student model to get higher accuracy, which in turn further improves the ensemble teacher.

## B PDED FRAMEWORK DETAILS

### B.1 ARZ MODEL

The second-order ARZ model introduces the acceleration of traffic flow as a state variable:

$$
\begin{aligned}
&\rho_t + (\rho v)_x = 0, \\
&(v + h(\rho))_t + v(v + h(\rho))_x = (U_{eq}(\rho) - u)/\tau, \\
&h(\rho) = U_{eq}(0) - U_{eq}(\rho), \\
&U_{eq} = v_f(1 - \frac{\rho}{\rho_m}).
\end{aligned}
\tag{33}
$$

where $h(\rho)$ is traffic pressure, $U_{eq}(\rho)$ is the equilibrium velocity, $v_f$ and $\rho_m$ are the maximum velocity and density, respectively.

The loss functions of the ARZ model is calculated as follows:

$$
\begin{aligned}
&\mathcal{L}_{pde-arz} = \frac{1}{N_p} \sum_{i=1}^{N_p} \left| f_{arz_1}(x^i, t^i, \theta_p) \right|^2 + \left| f_{arz_2}(x^i, t^i, \theta_p) \right|^2, \\
&f_{arz_1}(x, t, \theta_p) = (\rho_p)_t + (\rho_p v)_x, \\
&f_{arz_2}(x, t, \theta_p) = (v + h(\rho_p))_t + v(v + h(\rho_p))_x - (U_{eq}(\rho_p) - u)/\tau.
\end{aligned}
\tag{34}
$$

where $f(x, t, \theta_p)$ denotes the residual value of PDE, $N_p$ is the number of auxiliary points. Eq.(34) corresponds to the loss functions of the ARZ model. By minimizing the loss of Eq.(34), the optimal solution for the physics-informed model can be obtained based on the ARZ model.

## B.2 The Calculation of Traffic State Relation Metrics

First, let

$$\tilde{\mathbf{G}}_t = \mathbf{E}_{t_1} \cdot \mathbf{E}_{t_1}^\top; \mathbf{G}_{t[i,:]} = \tilde{\mathbf{G}}_{t[i,:]}/||\tilde{\mathbf{G}}_{t[i,:]}||_2 \tag{35}$$

where $\mathbf{E}_{t_1} \in \mathbb{R}^{N_d \times d}$ is the first layer output of teacher model, $N_d$ is the number of samples and $d$ is the dimension of representation, $\tilde{\mathbf{G}}_t$ is a $N_d \times N_d$ matrix. Then we apply a row-wise L2 normalization to obtain the normalized $\mathbf{G}_t$. Analogously, let

$$\tilde{\mathbf{G}}_s = \mathbf{E}_{d_2} \cdot \mathbf{E}_{d_2}^\top; \mathbf{G}_{s[i,:]} = \tilde{\mathbf{G}}_{s[i,:]}/||\tilde{\mathbf{G}}_{s[i,:]}||_2 \tag{36}$$

where $\mathbf{E}_{d_2} \in \mathbb{R}^{N_d \times d}$ and $G_s$ is a $N_d \times N_d$ matrix. Similarly, we apply a row-wise L2 normalization to obtain the normalized $\mathbf{G}_s$.

## B.3 Training Procedure

We iteratively update both the teacher and the student models. Algorithm 1 is the pseudo-code of our PDED framework.

---

**Algorithm 1** The PDED framework

---

**Require:** labeled dataset $\mathcal{O} = \{(x_o^{(i)}, t_o^{(i)})\}_{i=1}^{N_d}$, learning rate of physics-informed model, data-driven model and ensemble teacher model are $\gamma_{\text{phy}}$, $\gamma_{\text{data}}$ and $\gamma_{\text{T}}$ respectively.
**Ensure:** Predict result $\hat{\rho}$.
 1: Initialize the parameter related to physics-informed model, data-driven model and ensemble teacher model as $\theta_{\text{p}}, \theta_{\text{d}}, \theta_{\text{t}}$;
 2: **repeat**
 3:     $t = t + 1$;
 4:     get representation $\mathbf{E}_{p_1}$ and $\mathbf{E}_{d_1}$, uncertainty weight $\lambda_d$;
 5:     get the ensemble teacher model input $\mathbf{E}_{\mathbf{t}_1}$ via Eq.( 8);
 6:     get $\rho_t$ via Eq.( 9);
 7:     Update $\theta_{\text{t}}$ through $\mathcal{L}_{mse-t}$ in Eq.( 10) with labeled dataset $\mathcal{O}$;

$$\theta_{\text{t}} \leftarrow \theta_{\text{t}} + \gamma_{\text{t}}\frac{\partial L_{\theta_t}}{\partial \theta_{\text{t}}} \tag{37}$$

 8:     get $\rho_p$ and $\rho_d$ via Eq.( 3) and Eq.( 6);
 9:     Update $\theta_{\text{p}}, \theta_{\text{d}}$ through Eq.( 14) and Eq.( 15);

$$\theta_{\text{p}} \leftarrow \theta_{\text{p}} + \gamma_{\text{p}}\frac{\partial L_{\theta_{\text{p}}}}{\partial \theta_{\text{p}}} \tag{38}$$

$$\theta_{\text{d}} \leftarrow \theta_{\text{d}} + \gamma_{\text{d}}\frac{\partial L_{\theta_{\text{d}}}}{\partial \theta_{\text{d}}} \tag{39}$$

10: **until** convergence

---

# C Supplementary experiments

## C.1 Dataset and Experiment Setting

**Dataset**: US-80 dataset is a 1600 feet long road segment with recorded vehicle trajectory in 15 minutes ($x, t \in [0, 1600] \times [0, 900]$). The spatial resolution of the dataset is 20 feet and the temporal resolution is 5 second ($\Delta x = 20$, $\Delta t = 5$). Thus, we achieve a traffic state field that consists of 180 temporal bins and 80 spatial bins for model training. Similarly, US-101 dataset is a 630 meters long road segment with recorded vehicle trajectory in 255 seconds ($x, t \in [0, 630] \times [0, 255]$). The spatial resolution of the dataset is 30 meters and the temporal resolution is 1.5 second ($\Delta x = 30$, $\Delta t = 1.5$). Thus, we achieve a traffic state field that consists of 170 temporal bins and 21 spatial bins for model training.

**Evaluation metric**: We use the $L^2$ relative error to quantify the estimation error for traffic state estimation.

$$Err(\rho, \rho_{true}) = \frac{\sqrt{\sum_{r=1}^{N_d} \left| \rho(t^{(r)}, x^{(r)}; \theta) - \rho_{true}(t^{(r)}, x^{(r)}) \right|^2}}{\sqrt{\sum_{r=1}^{N_d} \left| \rho_{true}(t^{(r)}, x^{(r)}) \right|^2}}. \tag{40}$$

The same calculation process of the $L^2$ relative error $Err(v, v_{ture})$ applies to velocity $v$.

## C.2 ADDITIONAL ABLATION STUDY

Here, we present the ablation results of PDED-ARZ with 30% data in Table 4, as well as the ablation results of two variants under 50% data conditions in Table 5 and Table 6. Under different experimental settings, each module of our PDED framework shows significant effectiveness. Overall, we can observe that the ensemble distillation framework plays a a vital role in improving the performance, since either removing the distillation loss in data-driven model or in the physics information model, the model performance has a significant decrease. Moreover, when both losses are removed, the model performance declines more in all different situations. In addition, the BNN module and relationships between traffic state variables are also beneficial to the overall performance. Furthermore, more data can effectively improve the traffic state estimation performance. Also, consistent with the conclusion obtained in the overall performance, both LWR model and its variants outperform the ARZ model and its variants for 30% data and 50% data.

Table 4: Ablation study of PDED-ARZ model with 30% data. (Error). Bold: best

| Models | US-101 | | US-80 | | Synthetic | |
|---|---|---|---|---|---|---|
| | $v$ | $\rho$ | $v$ | $\rho$ | $v$ | $\rho$ |
| w/o $\mathcal{L}_r$ | 0.0758 | 0.1947 | 0.1074 | 0.1910 | 0.1276 | 0.1133 |
| w/o BNN | 0.0791 | 0.1963 | 0.1158 | 0.1981 | 0.1291 | 0.1176 |
| w/o $\mathcal{L}_{mse-st}$ | 0.0828 | 0.2024 | 0.1283 | 0.2039 | 0.1328 | 0.1196 |
| w/o $\mathcal{L}_{pde-st}$ | 0.0864 | 0.2076 | 0.1247 | 0.2086 | 0.1352 | 0.1265 |
| w/o $\mathcal{L}_{mse-st} + \mathcal{L}_{pde-st}$ | 0.0897 | 0.2127 | 0.1453 | 0.2261 | 0.1453 | 0.1348 |
| PDED-ARZ | **0.0705** | **0.1932** | **0.1015** | **0.1902** | **0.1201** | **0.1065** |

Table 5: Ablation study of PDED-LWR model with 50% data. (Error). Bold: best

| Models | US-101 | | US-80 | | Synthetic | |
|---|---|---|---|---|---|---|
| | $v$ | $\rho$ | $v$ | $\rho$ | $v$ | $\rho$ |
| w/o $\mathcal{L}_r$ | 0.0477 | 0.1607 | 0.0912 | 0.1758 | 0.0987 | 0.0868 |
| w/o BNN | 0.0492 | 0.1655 | 0.0942 | 0.1835 | 0.1065 | 0.0883 |
| w/o $\mathcal{L}_{mse-st}$ | 0.0564 | 0.1724 | 0.0987 | 0.1874 | 0.1087 | 0.0934 |
| w/o $\mathcal{L}_{pde-st}$ | 0.0528 | 0.1788 | 0.1017 | 0.1904 | 0.1120 | 0.0981 |
| w/o $\mathcal{L}_{mse-st} + \mathcal{L}_{pde-st}$ | 0.0594 | 0.1866 | 0.1157 | 0.1998 | 0.1186 | 0.1037 |
| PDED-LWR | **0.0417** | **0.1572** | **0.0882** | **0.1717** | **0.0948** | **0.0821** |

## C.3 ADDITIONAL NOISE ROBUSTNESS ANALYSIS

We conduct experiments on a 30% penetration rate in the synthetic dataset. The noise data are generated by adding a white Gaussian noise $\epsilon \sim \mathcal{N}(0, 0.04)$ to the training synthetic data with different data noise ratios among {0.05, 0.1, 0.15, 0.2, 0.25}. According to Figure 4, our model performance is significantly better than the baseline models. As the noise ratio increases, the prediction error only slightly increases, which means the model performance decreases slowly, indicating our PDED framework has higher robustness against data noise compared to the NN and PIDL models.

Table 6: Ablation study of PDED-ARZ model with 50% data. (Error). Bold: best

| Models | US-101 | | US-80 | | Synthetic | |
|---|---|---|---|---|---|---|
| | $v$ | $\rho$ | $v$ | $\rho$ | $v$ | $\rho$ |
| w/o $\mathcal{L}_r$ | 0.0661 | 0.1879 | 0.0985 | 0.1944 | 0.1168 | 0.0881 |
| w/o BNN | 0.0672 | 0.1898 | 0.1048 | 0.1980 | 0.1197 | 0.0904 |
| w/o $\mathcal{L}_{mse-st}$ | 0.0705 | 0.1914 | 0.1124 | 0.2032 | 0.1253 | 0.0943 |
| w/o $\mathcal{L}_{pde-st}$ | 0.0688 | 0.1920 | 0.1084 | 0.2069 | 0.1284 | 0.1002 |
| w/o $\mathcal{L}_{mse-st} + \mathcal{L}_{pde-st}$ | 0.0754 | 0.2025 | 0.1256 | 0.2129 | 0.1311 | 0.1092 |
| PDED-ARZ | **0.0646** | **0.1875** | **0.0944** | **0.1877** | **0.1107** | **0.0845** |

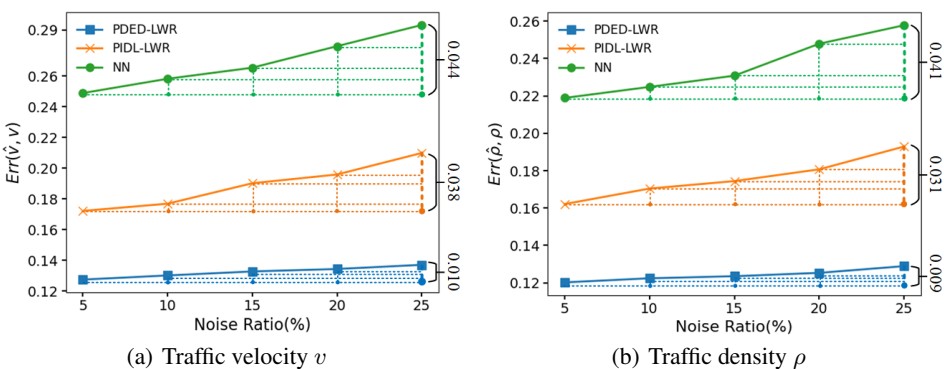

(a) Traffic velocity $v$        (b) Traffic density $\rho$

Figure 4: Noise robustness analysis for white Gaussian noise with variance 0.04. Similarly, PDED has much smaller increase of predcition error compared to the baselines.

## C.4 PARAMETERS SENSITIVITY ANALYSIS

In this section, we investigate the sensitivity of learning rates of the data-driven model, physics-informed model, and ensemble teacher model. The learning rates of the data-driven model and ensemble teacher model are set to be the same $lr$, varying among $\{0.01, 0.005, 0.001, 0.0005\}$. While the learning rate of physics-informed model $lr\_phy$ varies among $\{0.01, 0.005, 0.001, 0.0005. 0.0001\}$. We use PDED-LWR based on US-101 with 30% proportion of loop detectors (6L loop detectors) as an example to conduct parameters sensitivity analysis. We observe that: 1) For traffic velocity $v$ in Figure 5(a), as $lr$ increases, the model shows a trend of first decreasing and then increasing. The model ultimately achieves the optimal performance when $lr$ is 0.001 and $lr\_phy$ is 0.0001. 2) The traffic density $\rho$ demonstrates a different trend with $v$. As shown in Figure 5(b), with the increase of $lr$, the model performance shows a decreasing trend. While with the increase of $lr\_phy$, the model shows an upward trend. The model ultimately achieves the optimal performance when $lr$ is 0.01 and $lr\_phy$ is 0.005. Overall, when the proposed model gets the optimal performance, the $lr\_phy$ is much smaller than $lr$, since the loss of the physics-informed model is much smaller, and thus it needs a smaller learning rate.

## C.5 VISUALIZATION ANALYSIS

We visualize the predictions of the traffic velocity $v$ and traffic density $\rho$ by NN and proposed PDED with 30% data compared to the true values for the synthetic dataset, US-80, and US-101 in Figure 6, Figure 7, and Figure 8 respectively. We observe that in all scenarios, PDED extracts the significant characteristics of the synthetic data and NGSIM data, and reconstructs the different degrees of traffic velocity and density from limited data samples, which is highly close to the real situation. Moreover, PDED shows more clear boundaries among different traffic states. In contrast, no meaningful traffic insights are captured from the velocity and density fields estimated by NN, especially for US-101 dataset. The results demonstrate that the PDED achieves better performance from macroscopic perspective.

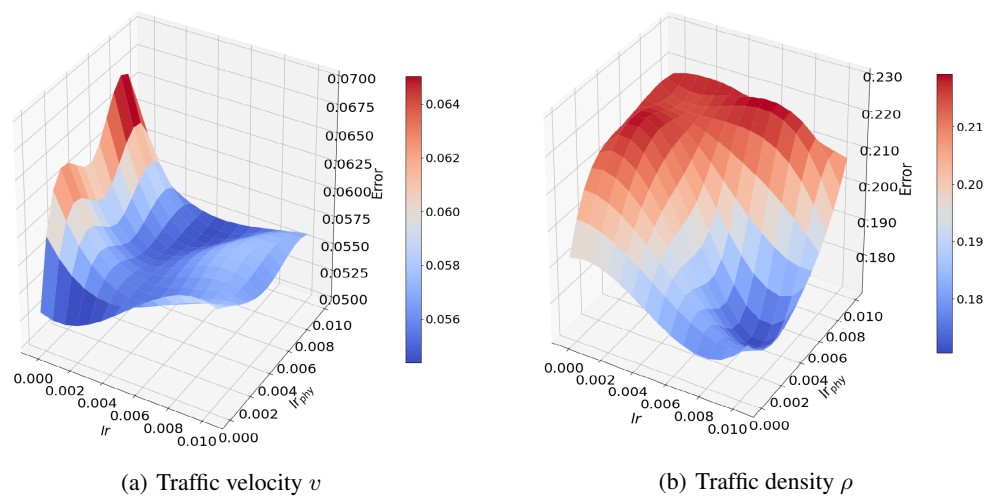

(a) Traffic velocity $v$            (b) Traffic density $\rho$

Figure 5: Parameters sensitivity analysis for US-101 with 30% proportion of loop detectors.

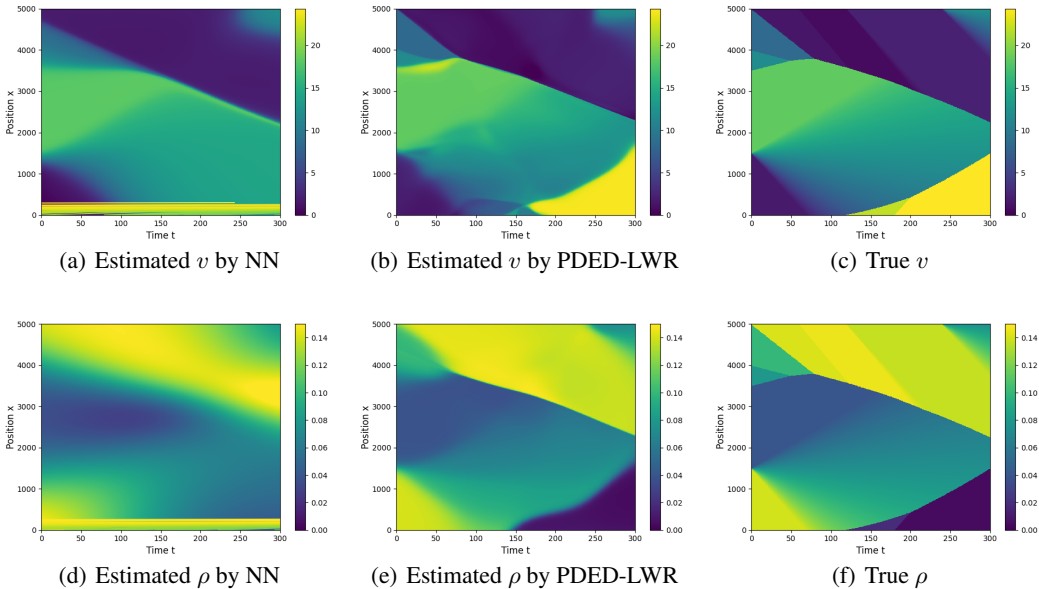

(a) Estimated $v$ by NN     (b) Estimated $v$ by PDED-LWR     (c) True $v$

(d) Estimated $\rho$ by NN     (e) Estimated $\rho$ by PDED-LWR     (f) True $\rho$

Figure 6: Visualization analysis of Synthetic dataset with 30% penetration rate.

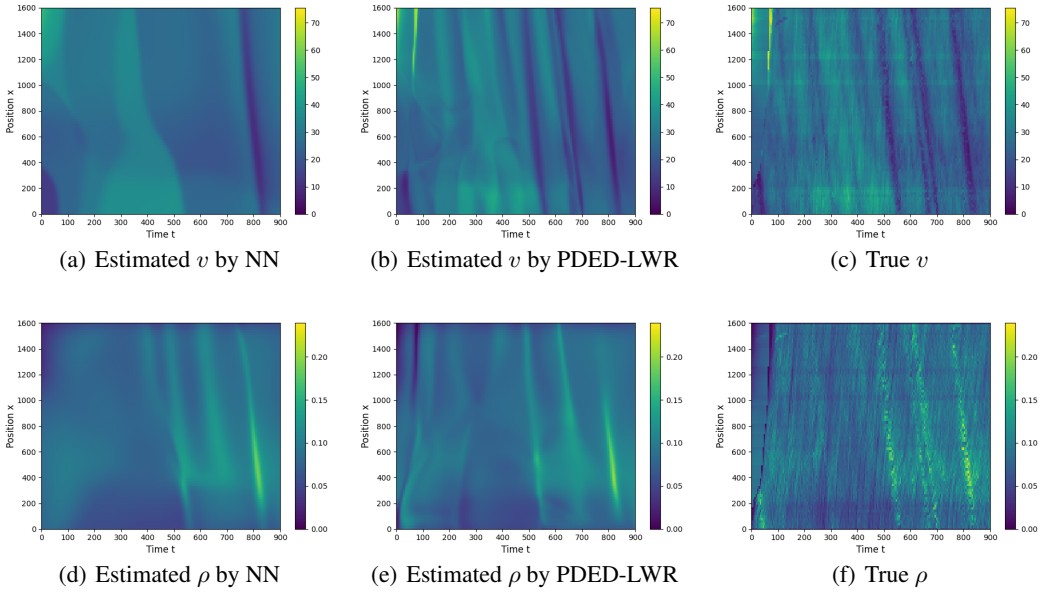

(a) Estimated $v$ by NN     (b) Estimated $v$ by PDED-LWR     (c) True $v$

(d) Estimated $\rho$ by NN     (e) Estimated $\rho$ by PDED-LWR     (f) True $\rho$

Figure 7: Visualization analysis of US-80 dataset with 30% proportion of loop detectors.

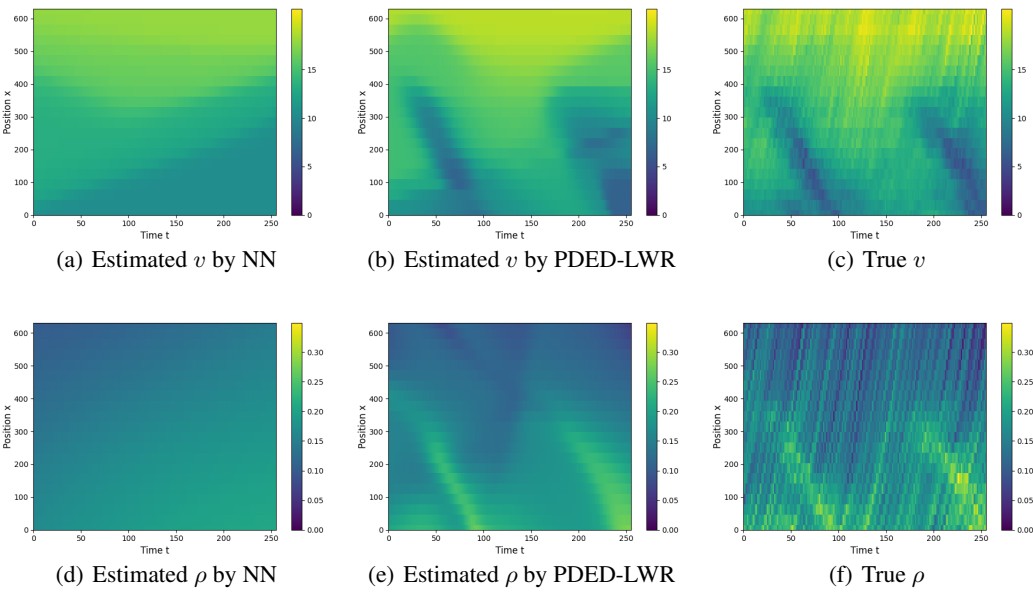

(a) Estimated $v$ by NN     (b) Estimated $v$ by PDED-LWR     (c) True $v$

(d) Estimated $\rho$ by NN     (e) Estimated $\rho$ by PDED-LWR     (f) True $\rho$

Figure 8: Visualization analysis of US-101 dataset with 30% proportion of loop detectors.

