# OpenReview forum: "PDED: Revitalize physics laws submerged in data information for Traffic State Estimation"
_ICLR.cc/2024/Conference — Submitted to ICLR 2024_

### Official Review · Reviewer_m66B · 2023-10-24

**Soundness:** 3 good
**Presentation:** 3 good
**Contribution:** 3 good
**Rating:** 5
**Confidence:** 4

**Summary:**

This paper proposes a new knowledge-guided ML method, called PDED. PDED disentangles the data-based and physics-based modules, which are trained with labeled data and physics laws respectively. Ensembling learning and knowledge distillation are used to assemble those two representations. Superior performance has been shown to validate the effectiveness of the proposed method.

**Strengths:**

- This paper proposes a new framework for combining physics laws and data, which aims to avoid the optimization issue in common physics-informed learning tasks.

- This paper is well-written and well-structured.

**Weaknesses:**

- It seems that this method still has many hyperparameters to tune as shown in Eq. (14) and (15), though the goal of this paper is to mitigate the optimization issue in physics-informed learning. I am not sure of the magnitude differences between those loss terms in physics loss and data loss in  Eq. (14) and (15).

- As shown in **Experiments settings** on Page 6, all of the hyper-parameters are set to 1. It seems they don’t have large variances in magnitudes. However, the test datasets focus on $x, t$ dimensions, which can be regarded as 1D PDEs in physics-informed learning. 1D PDEs are relatively easy to learn, and the optimization issues are not severe. I am wondering if the authors could test on more challenging 2D datasets in traffic modeling.

**Questions:**

- On Page 2, the authors claim that they discovered the optimization issue of physics-informed learning, where many existing research works have identified this problem [1-2]. I don’t think that is one of the contributions of this paper.

- On Page 5, “As mentioned by Theorem 2.1, the effectiveness of ensemble teacher model is also determined by the the accuracy of individual student model.” There is a double “the” in this sentence.

- This paper considers relative L2 errors, as commonly seen in physics-informed learning research. However, in the domain of traffic prediction, researchers also consider RMSE and MAPE to measure the errors from different perspectives. Is it possible to add more evaluation metrics since the traffic data has some rush hour phenomena, where extreme statistics are needed?

- Regarding the noise setting in Section 3.4, is it common to set the noise ratio by using a small covariance of 0.01/0.04 in the Gaussian distribution? Can the authors add some references here?

---
**Refs:**

[1] Krishnapriyan, Aditi, et al. "Characterizing possible failure modes in physics-informed neural networks." Advances in Neural Information Processing Systems 34 (2021): 26548-26560.

[2] Wang, Sifan, Yujun Teng, and Paris Perdikaris. "Understanding and mitigating gradient flow pathologies in physics-informed neural networks." SIAM Journal on Scientific Computing 43.5 (2021): A3055-A3081.

---

> ### Author Response · Authors · 2023-11-17
> **Reply to Reviewer m66B  Part 1**
>
> Thank you for your constructive comments and suggestions, and they are exceedingly helpful for us to improve our paper.
>
> Please see our responses below. We have started working on the corresponding changes, and will incorporate all of them in the final revision. If you have any further concerns, we would be keen to address them as well.
>
> **Q: I am not sure of the magnitude differences between those loss terms in physics loss and data loss in Eq. (14) and (15).**
>
> **A**: Sorry for the confusion. The magnitude differences exist between physics loss and data loss, as we proved in **Theorem A.1** in Appendix A.1. However, the magnitude differences may not exist if the losses are all physics losses, or all data losses. The losses $L_{pde}$ and $L_{pde-st}$ in Eq.(14) are both physics losses, while the losses $L_{mse-d}$, $L_{mse-st}$ and $L_r$ in Eq.(15) are all data losses. Thus, losses in Eq.(14) have relative same magnitude, and losses in Eq.(15) have relative same magnitude. Because we optimize Eq.(14) and Eq.(15) separately, the large magnitude differences between physics loss and data loss would not affect each other so that we can get the better performance. This is one of novelty for our model thanks to the idea of disentangling the data-driven model and physics-based model as students to fit their own objective functions.
>
> **Q: The question about Experiments settings on Page 6.**
>
> **A**:  Sorry for the confusion. As mentioned in above answers, the losses in Eq.(14) have relative same magnitude, while losses in Eq.(15) have relative same magnitude. Thus, we can set all of the hyper-parameters in Eq.(14) and Eq.(15) to 1 instead of spending lots of time to find proper hyper-parameters in existing methods. Actually, if the hyper-parameters in Eq.(14) and Eq.(15) are determined by a grid search during the training process, it may get better performance than the results shown in our paper. However, it seems not necessary since it will take a lot of time, and our model has already achieved better performance when they are set to 1 compared to the previous work.
>
> Moreover, the fact that the hyper-parameters in Eq.(14) and Eq.(15) are set to 1 has no direct relationship with whether it is 1D PDEs or 2D PDEs. As we disentangle the data-driven model and physics-based model as students to fit their own objective functions, they are optimized separately. So the hyper-parameters set to 1 is also applied to 2D PDEs. However, the existing methods just simply add physics loss and data loss, even for 1D PDEs which are relatively easy to learn, the magnitude differences between physics loss and data loss are huge (see left figure in Figure 1), and thus the hyper-parameters can not be set to 1. In conclusion, it is our framework of first disentangling and then assembling that makes the hyper-parameters in Eq.(14) and Eq.(15) set to 1 whether it is 1D PDE or 2D PDE, which strongly indicates the novelty of our framework.

---

> ### Author Response · Authors · 2023-11-17
> **Reply to Reviewer m66B Part 2**
>
> **Q: I don’t think that is one of the contributions of this paper.**
>
> **A**:  Sorry for the confusion. The first contribution is not finding the optimization issue of physics-informed learning, and it is the motivation to lead to our novel solution. We focus on the way to solve this problem that we propose to first disentangle and then assemble theory to manage physical knowledge and data information.
>
> **Q: There is a double “the” in this sentence.**
>
> **A**: Sorry for the mistake. We will delete one 'the' in the revised paper.
>
> **Q: Is it possible to add more evaluation metrics since the traffic data has some rush hour phenomena, where extreme statistics are needed.**
>
> **A**: Thank you for your suggestion. In fact, the reason we do not use other metrics is that almost all the existing methods for traffic state estimation only use the L2 error [1-5]. As you suggest to add more evaluation metric, we make efforts to add the RMSE for traffic state $\rho$ of our method and one best baseline for US-101. The results can be seen in the following table. It can be seen that the proposed method PDED has better performance. We will test RMSE for all baselines in our future work.
>
> | Loop     | 2      | 6      | 10     | 15     | 18     |
> | :------- | :----- | :----- | :----- | ------ | ------ |
> | PIDL-LWR | 0.0351 | 0.0307 | 0.0291 | 0.0281 | 0.0278 |
> | PDED-LWR | 0.0336 | 0.0287 | 0.0259 | 0.0236 | 0.0230 |
>
>
>
> **Q: Can the authors add some references here?**
>
> **A**: Thank you for your suggestion. It is common to set the noise ration by using a small covariance, please see the references [5-7].
>
>
>
> Refs:
>
> [1]Archie J Huang and Shaurya Agarwal. Physics informed deep learning for traffic state estimation.
> In 2020 IEEE 23rd International Conference on Intelligent Transportation Systems (ITSC), pp.
> 1–6. IEEE, 2020.
>
> [2]Rongye Shi, Zhaobin Mo, and Xuan Di. Physics-informed deep learning for traffic state estimation:
> A hybrid paradigm informed by second-order traffic models. In Proceedings of the AAAI Conference on Artificial Intelligence, volume 35, pp. 540–547, 2021a.
>
> [3]Rongye Shi, Zhaobin Mo, Kuang Huang, Xuan Di, and Qiang Du. Physics-informed deep learning
> for traffic state estimation. arXiv preprint arXiv:2101.06580, 2021b.
>
> [4]Archie J Huang and Shaurya Agarwal. Physics-informed deep learning for traffic state estimation:
> illustrations with lwr and ctm models. IEEE Open Journal of Intelligent Transportation Systems,
> 3:503–518, 2022.
>
> [5]Zhaobin Mo, Yongjie Fu, Daran Xu, and Xuan Di. Trafficflowgan: Physics-informed flow based
> generative adversarial network for uncertainty quantification. In Joint European Conference on
> Machine Learning and Knowledge Discovery in Databases, pp. 323–339. Springer, 2022.
>
> [6]Di X, Shi R, Mo Z, et al. Physics-Informed Deep Learning For Traffic State Estimation: A Survey and the Outlook. Algorithms, 2023, 16(6): 305.
>
> [7]Yang M, Foster J T. Multi-output physics-informed neural networks for forward and inverse PDE problems with uncertainties. Computer Methods in Applied Mechanics and Engineering, 2022, 402: 115041.

---

### Official Review · Reviewer_gduu · 2023-10-29

**Soundness:** 2 fair
**Presentation:** 3 good
**Contribution:** 1 poor
**Rating:** 3
**Confidence:** 2

**Summary:**

The paper proposes an ensemble (teacher-student) based physics-informed and data-driven model for traffic state estimation. Specifically, due to the well-known complexity of coupled modeling of physics information (in the form of PDE based conservation conditions) and data-driven (e.g., Mean-squared error) losses, the paper proposes a de-coupled method of modeling these two steps by first employing a physics-informed neural network to model the PDE based losses and a Bayesian Neural Network to model the data-driven losses. Further, a feed-forward neural network is employed to ensemble the predictions of the physics-informed and data-driven components by employing the latent representations from each of the two (I.e., physics-informed and data-driven) models. The ensemble model (infused with both data-driven and physics-driven knowledge) is trained in a data-driven manner and knowledge from the ensemble (teacher) model is distilled into each individual physics-driven, data-driven (student) models. In this manner, employing a combination of ensemble modeling, physics informed neural networks and knowledge distillation, the paper proposes a Physics-Informed Deep Learning (PIDL) solution for traffic state estimation.

**Strengths:**

1. Two interesting parts about the paper are (i) the alternating teacher / student training (ii) the uncertainty fusion employing the BNN covariance matrix.  However, not enough is discussed about either of these aspects in the main paper.


2. Overall the paper is cohesive and well-written.  The quantitative and qualitative results (although incomplete) demonstrate that the proposed framework yields good performance relative to baselines in addition to highlighting the importance of each component (ablation analysis).

**Weaknesses:**

- **[Limited Novelty].** The proposed model is a combination of a physics informed neural network (PINN) employed with a relatively simple PDE, in addition to another standard Bayesian Neural Network (BNN) paradigm combined with knowledge distillation (KD). These three paradigms (PINN, BNN, KD) are all extremely well studied, well-understood and the current paper doesn’t propose any novel extensions of the actual paradigms or increase the characteristic understanding of any of the aforementioned paradigms. It is simply an exercise in the application (specifically, the combination) of these paradigms in a (somewhat) creative way to address the problem of traffic state estimation.
    - Further, the knowledge that the physics losses and data-driven loss don’t always play well together (mentioned in contribution 1) isn’t new and has been well researched in the context of physics-informed neural networks applied to PDEs [3, 4, 5]


- **[Important Related Work Missing].** The paper completely misses mentioning operator learning paradigms which are more recent updates to traditional PINNs which learn families of PDEs as opposed to single instances of PDEs (albeit in a slightly different manner) however the reviewer believes that a brief description of operators like [1, 2] would contextualize the current work in the physics informed deep learning space.


- **[Incomplete Performance comparison].** A related paper Trafficflowgan [6] that employs physics information, uncertainty aware GAN for traffic state estimation has not been compared against.

### References:

1. Lu, Lu, Pengzhan Jin, and George Em Karniadakis. "Deeponet: Learning nonlinear operators for identifying differential equations based on the universal approximation theorem of operators." arXiv preprint arXiv:1910.03193 (2019).


2. Li, Zongyi, et al. "Fourier neural operator for parametric partial differential equations." arXiv preprint arXiv:2010.08895 (2020).


3. Krishnapriyan, Aditi, et al. "Characterizing possible failure modes in physics-informed neural networks." Advances in Neural Information Processing Systems 34 (2021): 26548-26560.


4. Wang, Sifan, Yujun Teng, and Paris Perdikaris. "Understanding and mitigating gradient flow pathologies in physics-informed neural networks." SIAM Journal on Scientific Computing 43.5 (2021): A3055-A3081.


5. Kim, Jungeun, et al. "DPM: A novel training method for physics-informed neural networks in extrapolation." Proceedings of the AAAI Conference on Artificial Intelligence. Vol. 35. No. 9. 2021.


6. Mo, Zhaobin, et al. "Trafficflowgan: Physics-informed flow based generative adversarial network for uncertainty quantification." Joint European Conference on Machine Learning and Knowledge Discovery in Databases. Cham: Springer Nature Switzerland, 2022.

**Questions:**

1. Why has TrafficflowGAN [6] not been compared despite being a related / physics-informed + uncertainty aware model for traffic state estimation?

---

> ### Author Response · Authors · 2023-11-17
> **Reply to Reviewer gduu**
>
> Thank you for your constructive comments and suggestions. They are exceedingly helpful for us to improve our paper.
>
> Please see our responses below. We have started working on the corresponding changes and will incorporate all of them in the final revision. If you have any further concerns, we would be keen to address them as well.
>
> **Q: Limited Novelty.**
>
> **A**: Thank you for your comments. Here is the novelty of the proposed model:
>
> 1. Though the problem that physics losses and data-driven loss don’t always play well together isn’t new and has been well researched in some papers, all their frameworks are the same that they still use the combination by adding the physics losses and data-driven loss together. We do some important experiments in Introduction Section to show that these simple integration way has many problems: large magnitude differences, varying convergence rates, and conflicting directions of the gradients so that the physics law can not work as expected. By finding these issues, we are motivated to propose our model to solve these problems.
> 2. Moreover, we theoretically analyze that the above issues will general exist in the previous methods in **Appendix A.1 and A.2**. The existing methods can not deal with the above problem simultaneously, since the framework of adding the physics losses and data-driven loss is the root problem. Thus, breaking the original integration way is urgent and innovative.
> 3. Therefore, the superiority of the proposed method is that we naturally propose an overall framework to disentangle the data-driven model and physics-based model as students to fit their own objective functions for the first time, whose rationale is analyzed in **Theorem 2.1**. Then we combine them in a more elegant way by assembling their embeddings to construct a competitive model. The key point is our whole framework where each parts have their own physical meanings. Our principle is utilizing concise and effective tools to construct the overall framework for better performance instead of designing the complex structure in each part, and the results show the superiority of our model.
>
> **Q: Important Related Work Missing.**
>
> **A**: Thank you for your suggestion. In fact, we focus on designing a model for a certain situation with a fixed boundary condition, thus we do not describe the operators method in related work. But we will follow your suggestion to give a brief review of operators in our revised paper.
>
> **Q: Incomplete Performance comparison.**
>
> **A**: Thank you for your suggestion. We have noticed the paper TrafficFlowGAN, and we originally intended to compare with it. There are the following reasons that we do not compare with TrafficFlowGAN:
>
> 1. The results summarized in their paper are shown in figures instead of tables. Thus, we do not know the exact metric values for different situations.
>
> 2. Because our experiment setting has changed, we need to rerun the codes of TrafficFlowGAN to make fair comparison. However, we can not get the results with the given parameters by authors for 20000 epochs, since it results in gradient explosion as mentioned in their paper, though the author has made some improvement.
>
> As you mention the TrafficFlowGAN, we make efforts to achieve it.  Since it may occur gradient explosion and it takes a long time to run, we can only give the best performance of TrafficFlowGAN for 8000 epochs with the parameters the author provided for US-101 dataset in limited time. It can be seen that the results have lower performance than our model.
>
> |                |               | Loop   |        |        |
> | :------------- | :------------ | :----- | :----- | ------ |
> | Model          | Traffic State | 2      | 6      | 10     |
> | TrafficFlowGAN| v             | 0.1925 | 0.1851 | 0.1757 |
> | PDED           | v             | 0.1237 | 0.0531 | 0.0417 |
> | TrafficFlowGAN| $ \rho $        | 0.3095 | 0.2858 | 0.2547 |
> | PDED           | $ \rho $       | 0.2045 | 0.1724 | 0.1572 |

---

> > ### Comment · Reviewer_gduu · 2023-11-21
> > **Response to Authors**
> >
> > a. "We do some important experiments in Introduction Section to show that these simple integration way has many problems: large magnitude differences, varying convergence rates, and conflicting directions of the gradients so that the physics law can not work as expected."
> >
> > __Response1__
> > >There have been many papers (2 examples below) that have demonstrated this previously in the Physics Informed Machine Learning realm (e.g., check Fig.2 in paper [1] cited below, Fig. 2 in paper [2] cited below), so I still am not convinced that this can be considered a contribution.
> > >### References:
> > >1. Kim, Jungeun, et al. "DPM: A novel training method for physics-informed neural networks in extrapolation." Proceedings of the AAAI Conference on Artificial Intelligence. Vol. 35. No. 9. 2021.
> > >2. Wang, Sifan, Yujun Teng, and Paris Perdikaris. "Understanding and mitigating gradient flow pathologies in physics-informed neural networks." SIAM Journal on Scientific Computing 43.5 (2021): A3055-A3081.
> >
> >
> > b. Incomplete Performance Comparison
> >
> > __Response 2__
> > >Considering the `Experimental Settings` described in the paper, it can be gleaned that there have been multiple hyperparameter settings searched over before reporting results of the PDED model (at least as understood by the reviewer). However, you explanation seems to suggest that this was not carried out for the `TrafficFlowGAN` . In light of this, I don't think the results presented with the baseline model are too informative to understand where the proposed model stands with respect to previous state-of-the-art in the context of physics-informed traffic forecasting models. If possible, please present a more _representative_ comparison between the two models.

---

> > > ### Author Response · Authors · 2023-11-22
> > > **Reply to Reviewer gduu**
> > >
> > > **Q:  I still am not convinced that this can be considered a contribution.**
> > >
> > > **A**: Thank you for your comments. Though some papers have demonstrated these problems previously, they each have their own limitations. Take DPM [1] and Reference [2] which you mentioned for examples:
> > >
> > > 1.	DPM [1] finds that physics loss and data-driven loss have different convergence rates and conflicting directions and proposes to dynamically manipulate the gradients. However, it still has the problems that: 1) DPM does not have any theoretical analysis; 2) DPM ignores the problem that there is large magnitude differences between different losses.
> > >
> > > 2.	Reference [2] finds the unbalanced back-propagated gradients during model training by experiment and theoretical proof, and proposes a learning rate annealing algorithm to solve this problem. But it can not deal with different convergence rates and conflicting directions of the gradients.
> > >
> > > However, we introduce the problems in a more **systematic and complete way**. Also, we conduct relatively **comprehensive experiments and theoretical proofs**, including those they did not mention, such as our paper uses theory to prove that there is a conflict between the gradient directions of physics loss and data-driven loss.
> > >
> > > On the other hand, the above aspects are **the motivation to our new solution**. Previous papers have made improvements to gradient based on traditional PIDL framework, which has limitations mentioned in our paper and cannot solve the problem of magnitude, convergence rates and directions simultaneously. Different from their methods, we propose to **first disentangle and then assemble theory** to manage physical knowledge and data information, which can solve the entire problems.
> > >
> > > Of course, although these papers differ from our research, we still appreciate mentioning them and we will add citations as well as discuss the difference in detail in the final version.
> > >
> > > **Q:  Incomplete Performance Comparison.**
> > >
> > > **A**: Thank you for your suggestion. As you suggest, we fine-tune the parameters of TrafficFlowGAN again. Since it may result in gradient explosion as mentioned in their paper, we give the best performance of TrafficFlowGAN for 8000 epochs. Their source code and data are available at https://github.com/ZhaobinMo/TrafficFlowGAN, and **you can also rerun the code if you think it is not a representative comparison between the two models.**
> > >
> > > |                |               | Loop   |        |        |
> > > | :------------- | :------------ | :----- | :----- | ------ |
> > > | Model          | Traffic State | 2      | 6      | 10     |
> > > | TrafficFlowGAN | v             | 0.1719 | 0.1621 | 0.1573 |
> > > | PDED           | v             | 0.1237 | 0.0531 | 0.0417 |
> > > | TrafficFlowGAN | $\rho$        | 0.2884 | 0.2631 | 0.2462 |
> > > | PDED           | $\rho$        | 0.2045 | 0.1724 | 0.1572 |

---

### Official Review · Reviewer_ntPW · 2023-10-30

**Soundness:** 3 good
**Presentation:** 4 excellent
**Contribution:** 3 good
**Rating:** 5
**Confidence:** 4

**Summary:**

The manuscript elucidates inherent limitations in conventional physics-informed deep learning, notably the diminution of physics insights due to disparate magnitudes, antithetical gradient orientations, and inconsistent convergence tempos between distinct loss functions. To address these challenges, the authors propose the Physical knowledge combined Data information neural network with Ensemble Distillation framework (PDED). This approach disentangles the data-driven and physics-based models, then reassembles them using ensemble learning and knowledge distillation. By incorporating Bayesian Neural Networks, they also manage data uncertainty.

**Strengths:**

1.The manuscript highlights the inherent limitations of the physics-informed deep learning approach. Especially, there are discrepancies in magnitude between the Loss_PDE and Loss_MSE, along with distinct convergence speeds and conflicting gradient orientations. Appendix A and Figure 1 provide a lucid exposition of these issues, serving as the major highlights of this paper.
2.PDED uses knowledge distillation, especially the ensemble distillation framework, to solve the conflict between Physics-Based model and Data-Driven model.
3. The manuscript provides a very thorough and comprehensive ablation study. Comprehensive mathematical demonstration is contained with details

**Weaknesses:**

1. This paper identifies a valuable research question, but its proposed solution looks a stack of existing ensemble distillation, traffic state relation, and anomaly detection via BNN into two models, stacking sufficient external strategies to boost performance.
2. Using Traffic State Relation Distillation, Ensemble Distillation, and anomaly detection via BNN lacks motivation. The author's approach seems to amalgamate all available techniques into the model to ascertain if performance can be improved, resembling a combination of existing strategies. At least within this paper, the author fails to provide a rationale for choosing this particular strategy, thereby giving readers an impression of a brute-force stacking of various established methods.
3. Experiment 3.4 Noise Robustness Analysis
The PDED has better noise robustness compared to the baselines since it utilizes the BNN to measure data uncertainty and adopts uncertainty to guide the fusion process. However, the comparison with the other two models is not rigorous enough. While a significant part of the robustness improvement in your model can be attributed to the integration of the BNN, it's inconclusive whether the NN and PIDL models, if augmented with uncertainty quantification, would demonstrate weaker noise robustness than PDED.

**Questions:**

1.	Could you elaborate on the difference/superiority between the proposed method and existing works on combining physical losses and standard ML losses (e.g., MSE)?

2.	See other comments in weaknesses above

---

> ### Author Response · Authors · 2023-11-17
> **Reply to Reviewer ntPW**
>
> Thank you for your constructive comments and suggestions. They are exceedingly helpful for us to improve our paper.
>
> Please see our responses below. We have started working on the corresponding changes and will incorporate all of them in the final revision. If you have any further concerns, we would be keen to address them as well.
>
> **Q: Could you elaborate on the difference/superiority between the proposed method and existing works on combining physical losses and standard ML losses (e.g., MSE)?**
>
> **A**: Sorry for the confusion. Here are the motivation and superiority of the proposed method:
> 1. We do some important experiments in Introduction Section to show that the existing works on adding physical losses and standard MSE losses have many problems: large magnitude differences, varying convergence rates, and conflicting directions of the gradients so that the physics law can not work as expected. By finding these issues, we are motivated to propose our model to solve these problems.
> 2. Moreover, we theoretically analyze that the above issues will general exist in the previous methods in **Appendix A.1 and A.2**. The existing methods can not deal with the above problem simultaneously, since the framework of adding the physics losses and data-driven loss is the root problem. Thus, breaking the original integration way is urgent and innovative.
> 3. Therefore, the superiority of the proposed method is that we naturally propose an overall framework to disentangle the data-driven model and physics-based model as students to fit their own objective functions for the first time, whose rationale is analyzed in **Theorem 2.1**. Then we combine them in a more elegant way by assembling their embeddings to construct a competitive model. The key point is our whole framework where each parts have their own physical meanings. Our principle is utilizing concise and effective tools to construct the overall framework for better performance, and the results show the superiority of our model, which is not a brute-force stacking of various established methods.
>
>  **Q: It's inconclusive whether the NN and PIDL models, if augmented with uncertainty quantification, would demonstrate weaker noise robustness than PDED.**
>
> **A**: Sorry for the confusion. We add more experiments to show that our model has better noise robustness not only because of the BNN module but also other parts such as assembling features of student models and distilling knowledge from teacher model.  PDED-LWR w/o BNN denotes that PDED use NN instead of BNN in data-driven model. The results of L2 relative error for traffic state v on a 30% penetration rate in the synthetic dataset with different data noise ratios (Gaussian noise with variance 0.01) are shown as follows:
>
> | Noise Ratio      | 5%     | 10%    | 15%    | 20%    | 25%    |
> | :--------------- | :----- | :----- | :----- | :----- | :----- |
> | NN               | 0.1943 | 0.1996 | 0.2043 | 0.2143 | 0.2234 |
> | PIDL-LWR         | 0.1612 | 0.1644 | 0.1709 | 0.1767 | 0.1844 |
> | PDED-LWR w/o BNN | 0.1365 | 0.1387 | 0.1421 | 0.1466 | 0.1482 |
> | PDED-LWR         | 0.1205 | 0.1228 | 0.1247 | 0.1259 | 0.1275 |
>
> As shown in above table, we can see that even without BNN module, the prediction error of our framework also increases slowly than NN and PIDL models, which indicates that our model has better noise robustness not only because of using the BNN module. It also proves the novelty of our framework.

---

### Meta-Review · Area_Chair_5buT · 2023-12-09

**Metareview:**

This paper proposes PDED, an ensemble method, to combine physics-informed and data-driving components for traffic state estimation. The reviewers generally agree that this paper is well-structured and well-written. Two out of three reviewers have concerns on the novelty and intuition of the method, where the reviewers think that PDED is a combination of three different methods (PINN, BNN, KD) that are well studied, and the result is a straightforward application. There is also concerns on lacking experiment comparison with a recent (Yr 2022) baseline Trafficflowgan, and the authors have managed to demonstrate simulation results with better performance. However, one reviewer is still not convinced and it is not clear if this concern has been resolved.

Based on the reviewers rating and rebuttal discussion, I recommend rejection and sincerely encourage the authors to revise the paper to incorporate all the reviewers feedback, which will strengthen the submission for future venue.

**Justification For Why Not Higher Score:**

recommending reject due to novelty and experiment concerns.

**Justification For Why Not Lower Score:**

N/A

---

### Decision · Program_Chairs · 2024-01-16

Reject